# The pleasurable urge to move to music is unchanged in people with musical anhedonia

**Isaac D. Romkey**[1,2,3]*, **Tomas Matthews**[4,5], **Nicholas Foster**[2,3,6], **Simone Dalla Bella**[2,3,6,7], **Virginia B. Penhune**[1,2,3]

**1** Department of Psychology, Concordia University, Montréal, Quebec, Canada, **2** International Laboratory for Brain, Music and Sound Research (BRAMS), Montréal, Quebec, Canada, **3** Center for Research in Brain, Language and Music (CRBLM), Montréal, Quebec, Canada, **4** Center for Music in the Brain, Department of Clinical Medicine, Aarhus University Hospital, Aarhus, Denmark, **5** Royal Academy of Music, Aarhus C, Denmark, **6** Department of Psychology, University of Montréal, Montréal, Quebec, Canada, **7** University of Economics and Human Sciences in Warsaw, Warsaw, Poland

* Isaac.romkey@mail.concordia.ca

**Data Availability Statement:** All data that support these findings are publicly available on the Open Science Framework at the following link: https://DOI.org/10.17605/OSF.IO/GPQEB.

## Abstract

In cognitive science, the sensation of "groove" has been defined as the pleasurable urge to move to music. When listeners rate rhythmic stimuli on derived pleasure and urge to move, ratings on these dimensions are highly correlated. However, recent behavioural and brain imaging work has shown that these two components may be separable. To examine this potential separability, our study investigates the sensation of groove in people with specific musical anhedonia. Individuals with musical anhedonia have a blunted ability to derive pleasure from music but can still derive pleasure from other domains (e.g., sex and food). People with musical anhedonia were identified as those with scores in the lower 10% of scores on the Barcelona Musical Reward Questionnaire, but who had no deficits in music perception, no symptoms of depression, average levels of physical and social anhedonia, and sensitivity to punishment and reward. We predicted that if the two components of groove are separable, individuals with musical anhedonia would experience lower levels of derived pleasure but have comparable ratings of wanting to move compared to controls. Groove responses were measured in an online study (N = 148) using a set of experimenter-generated musical stimuli varying in rhythmic and harmonic complexity, which were validated in several previous studies. Surprisingly, we found no significant differences in groove response between individuals with musical anhedonia (n = 17) and a matched control group (n = 17). Mediation analyses for the anhedonia sample found that wanting to move ratings fully mediated the effect of rhythmic and harmonic complexity on pleasure ratings. Taken together, these results indicate that the urge to move may compensate for the blunted pleasure sensation in those with musical anhedonia. More generally, these results suggest that the urge to move is a primary source of pleasure in the groove response.

**Funding:** This research was supported by a grant to VP from the Natural Sciences and Engineering Research Council of Canada (NSERC 2021-04026). IR was funded by the Canadian Graduate Scholarship - Masters by NSERC and by the Fonds de Rechereche du Québec - Nature et Technologies (FRQ - NT). SDB and NF received funding from a Discovery Grant (RGPIN-2019-05453) from NSERC, and by the Canada Research Chair program (CRC in music auditory-motor skill learning and new technologies). The funders had no role in study design, data collection and analysis, decision to publish, or preparation of the manuscript.

**Competing interests:** SDB is on the board of the BeatHealth company dedicated to the design and commercialization of technological tools for assessing rhythm capacities such as BAASTA and implementing rhythm-based interventions. This does not alter our adherence to PLOS ONE policies on sharing data and materials.

## Introduction

For most of history, humans have listened to and created music [1]. We use music for various reasons, including mood regulation, social connection, and dancing [2–4]. One of the more intriguing features of listening to music is that it is often accompanied by a pleasurable urge to move along. In music cognition literature, this pleasurable urge to move to music has been termed "groove" [5, 6]. The experience of groove is often described as having two different components, measured through ratings of pleasure and wanting to move, which are typically highly correlated ($r > 0.8$) [7–10]. The strength of this relationship may depend on if wanting to move and pleasure are rated directly after one another or not [11]. A recent review has begun to refer to the two as one sensation, rather than two separate components of a perception [12]. However, recent behavioural and neuroimaging work suggests that these components may be at least partially separable [8, 11]. Supporting this line of thinking, analysis of responses to a groove questionnaire found two separable factors, pleasure and urge to move, that were highly correlated [13]. Additionally, Senn and colleagues proposed a model for groove in 2019, in which the urge to move is placed as the central experience of groove [14]. These findings raise the question of whether groove is a joint response—with pleasure and urge to move more intertwined—or if they are separable components of the sensation. To test the separability of these components of the groove sensation, we assessed pleasure and wanting to move ratings in people with specific musical anhedonia who have a blunted pleasure response to music [4]. We hypothesized that if the experience of pleasure and wanting to move were separable, people with musical anhedonia would show a blunted pleasure response but a preserved desire to move.

The experience of groove is related to various structural components of music influence, including meter [6, 15], syncopation [7–9, 11], harmonic complexity [8, 9, 11], and style [5, 6]. Most of the work has focused on the rhythmic attributes of music, particularly syncopation. The term syncopation refers to a violation of the internal pulse or meter (i.e., the regularly recurring rhythmic pattern in the music) of a piece of music [16]. Syncopation has been found to have a quadratic, inverted-U-shaped relationship with ratings of pleasure and wanting to move, such that medium levels of syncopation result in the highest ratings, and both lower and higher levels of syncopation result in lower ratings [7]. One useful theory about why groove has an inverted-U shaped relationship with syncopation is predictive coding [12, 17–19]. The theory proposes that when we listen to music, there are two parallel processes resulting in our sensation of groove: one bottom-up process where we perceive the sequence of notes and their temporal components, and a second top-down process based on an internal model or a set of expectations and predictions about musical structure. According to this model, music-induced pleasure arises from the intrinsic reward associated with a balance between predictability and uncertainty [12, 17–19]. When we listen, we compare the incoming sensory information to our top-down model, and then update the model in response to any differences. Moderately complex rhythms offer both predictability and some surprise, allowing our internal model to update, and providing pleasure through the process of learning [12, 17–19]. Contrary to this, simple rhythms provide more predictability, but there is not enough surprise for the model to update. Meanwhile, in complex rhythms, the inverse is true: there is too much surprise and not enough predictability for the errors to be informative. The urge to move is hypothesized to come about when the rhythmic structure does not match our top-down model, and the brain attempts to reduce the error in predictions by engaging real or imagined movements [12, 17–19]. Further, these real or imagined movements allow us to assess the errors in our predictions that then lead to us improving our motor model (i.e., learning) and providing pleasure. The quadratic effect for rhythmic complexity has been replicated across a variety of music stimuli,

including natural excerpts and stimuli composed to fit specific experimental parameters [7–9, 15, 20]. This being said, two recent studies by Senn and colleagues found that syncopation and groove ratings did not share a quadratic relationship; one found a small positive linear effect [10], and the other found no effect at all [21].

Syncopation is not the only structural component of music that influences our groove response. In 2019, Matthews and colleagues explored the role of harmonic complexity in the pleasurable urge to move to music [8]. Harmonic complexity was manipulated by altering the content of the chords used to mark the rhythm. The authors found that low and medium harmonic complexity generated the highest levels of groove response compared to high harmonic complexity. Mediation analyses were also conducted to determine if either pleasure or wanting to move ratings were the driving factor in the relationship between harmonic complexity and groove. The analyses showed that pleasure ratings fully mediated the relationship between wanting to move ratings and harmonic complexity, indicating that harmonic complexity did not affect wanting to move directly, and only through its relationship with pleasure. These results support the possibility that the pleasure and wanting to move components of the groove sensation may be at least partially separable.

Additional evidence comes from a functional magnetic resonance imaging (fMRI) study from our laboratory [11]. The results showed that pleasure ratings were more associated with activity in reward networks, such as the right nucleus accumbens (NAcc) [22]; while wanting to move ratings were more associated with motor networks, such as the putamen and the caudate [23]. These results further support the idea that the pleasure and urge to move experienced in the groove response may be separable.

In order to further explore the separability of pleasure and wanting to move, we decided to assess the groove response in a large control sample and in people with specific musical anhedonia. While we often think of enjoying music as a universal experience, this is not true for everyone. Some individuals have a blunted ability to derive pleasure from music but are still able to derive pleasure from other aspects of life, such as sex or food [4]. This specific blunted pleasure response has been termed musical anhedonia. The fact that people experience anhedonia in one domain and not in others implies that pleasure can arise from various sources, and we have distinct hedonic responses for each source. Presently, there is no consensus on specific brain areas associated with musical anhedonia [24]. More recently, musical anhedonia has begun to be conceptualized as a disorder of connectivity [25]. In support of this, previous work has found that those with musical anhedonia have disrupted functional connectivity between the primary auditory cortex and reward networks (ventral striatum) [26]. If pleasure and urge to move are separable, then those with musical anhedonia should show reduced ratings for pleasure and relatively preserved ratings for urge to move. If pleasure and the urge to move are not separable process, we should see a reduction in ratings for both in those with musical anhedonia. If the control and musical anhedonia groups do not differ from one another, we will conduct mediation analyses to explore the possibility that in those with musical anhedonia, the urge to move preserves the pleasure response.

## Methods

The protocol was approved by the Concordia University Office of Research, Research Ethics Unit (#30004370). The present study was conducted in accordance with the local legislation and institutional requirements. Participants were recruited through Prolific and tested online through the Pavlovia platform. Prior to beginning the study protocol, participants provided written informed consent. Participants were provided monetary compensation in the form of 7.5 euros per hour. Data was collected from February 9th to July 29th, 2022.

## Participants

A total of 204 participants were tested. Twenty-five participants were excluded for failing to follow instructions and/or for invalid responding (e.g., three or more questionnaires in which their responses did not show appropriate variation). An additional 12 participants were excluded for having Amusia as determined by Montreal Battery for Amusia (MBEA) [27]. Seventeen individuals were identified as having specific musical anhedonia based on the criteria described below (Age Range = 21–67, Age $M$ = 33.12, Age $SD$ = 13.87, Females = 4, Males = 13, Years of Musical Training $M$ = 4.29, Years of Musical Training $SD$ = 11.03). In this study, we had two control samples. One which included all participants who did not meet criteria for musical anhedonia ($n$ = 148, Age Range: 18–67, Age $M$ = 26.05, Age $SD$ = 7.76, Females = 75, Males = 71, Years of Musical Training $M$ = 4.42, Years of Musical Training $SD$ = 6.3) and the other that was matched to the musical anhedonia group based on age, sex, and years of musical experience ($n$ = 17, Age Range = 22–63, Age $M$ = 33.06, Age $SD$ = 10.58, Females = 4, Males = 13, Years of Musical Training $M$ = 4.42, Years of Musical Training $SD$ = 6.3).

## Identification of specific musical anhedonia

Musical anhedonia refers to a specific blunting of the pleasure response to music that is not better explained by an impairment in perception, embedded in a more global anhedonia or depressive symptomology [4]. In order to identify individuals with specific musical anhedonia, we used the BMRQ [28], as done in previous work [26–29]. Individuals with musical anhedonia have previously been identified as individuals who score in the bottom 10% of total scores (65 or lower) [4, 26]. In addition to the BMRQ, participants completed measures assessing various domains to determine that their anhedonia was specific to music.

To rule out the presence of anhedonia secondary to depression, we had participants complete the Beck Depression Inventory-II (BDI-II) [30]. A symptom of depressive disorders are feelings of global anhedonia [31]. The BDI-II is a 21-item questionnaire, where for each item, participants are required to select one of four statements that best describes themselves over the past two weeks. The BDI-II is considered a gold-standard self-report measure of depressive symptomology [32, 33]. Participants who met criteria for moderate to severe levels of depressive symptoms were not considered to have specific musical anhedonia.

While certain conditions are characterized by anhedonia, it is also possible that individuals may also have varying levels of anhedonia from other sources. To rule out global anhedonia, we employed the Anhedonia Scales from the Wisconsin Schizotypal Scales, henceforth referred to as the Wisconsin Anhedonia Scales (WAS) [34, 35]. To rule out blunted global sensitivity to punishment and reward, participants were asked to complete the sensitivity to punishment and reward questionnaire (SPSRQ) [36]. Another way to conceptualize individuals' sensitivity to reward and punishment is to think of it in terms of motivational systems. We wanted to assess participants' motivation to approach rewarding and avoid punishing stimuli, commonly referred to as the Behavioural Approach System (BAS) and the Behavioural Inhibition System (BIS) [37]. To quantify this, participants completed the BIS/BAS Scale [38]. Participants who scored outside of two standard deviations from the normative sample for the WAS, SPSRQ, and BIS/BAS were considered to not meet criteria for specific musical anhedonia following an adapted procedure based on Martinez-Molina et al. (2016) [26].

To determine if participants' abilities to perceive and produce rhythm fall within the average range, participants completed four subtests from the Battery for the Assessment of Auditory Sensorimotor and Timing Abilities (BAASTA) [39], adapted in an online version on BRAMS Online Testing Platform. To evaluate participants' ability to perceive rhythm,

**Table 1. Results from WAS, SPSRQ, BIS/BAS, & BAASTA for control and musical anhedonia sample.**

| | Full Control | | Musical Anhedonia | | Matched Control | |
|---|---|---|---|---|---|---|
| | M | SD | M | SD | M | SD |
| WAS: Physical Anhedonia | 10.30 | 1.54 | 9.65 | 2.12 | 10.5 | 1.55 |
| WAS: Social Anhedonia | 7.73 | 1.86 | 7.59 | 2.15 | 7.94 | 2.21 |
| BIS | 21.46 | 3.87 | 20.35 | 4.15 | 21.69 | 3.63 |
| BAS: Drive | 10.43 | 2.58 | 8.29 | 1.83 | 10.81 | 2.66 |
| BAS: Fun Seeking | 11.66 | 2.46 | 10.06 | 1.95 | 10.88 | 2.45 |
| BAS: Reward Responsiveness | 16.71 | 2.92 | 15.47 | 2.03 | 16.75 | 1.98 |
| SPSRQ: Sensitivity to Reward | 11.23 | 4.02 | 8.88 | 3.74 | 11.19 | 5.65 |
| SPSRQ: Sensitivity to Punishment | 15.7 | 4.7 | 15.12 | 4.7 | 12.75 | 6.69 |
| BAT: D-Prime | 2.55 | 0.9 | 2.81 | 0.73 | 2.5 | 0.78 |
| Un-paced Tapping: ITI CV | 0.068 | 0.034 | 0.058 | 0.015 | 0.070 | 0.046 |
| Paced Tapping Isochronous: VL | 2.67 | 1.07 | 2.6 | 0.55 | 3.22 | 0.79 |
| Paced Tapping to Music: VL | 1.7 | 1.94 | 1.43 | 1.8 | 1.93 | 1.97 |

WAS = Wisconsin Anhedonia Scales, BAS = Behavioural Activation Scales, BIS = Behavioural Inhibition Scales, SPSRQ = Sensitivity to Punishment and Reward Questionnaire, BAT = Beat Alignment Task, ITI = Inter-Tapping Interval, CV = Coefficient of Variation, VL = Logit-transformed Vector Length, M = mean, SD = standard deviation.

participants completed the Beat Alignment Test (BAT). To assess beat production abilities, participants completed three different tasks from the BAASTA; the un-paced tapping, paced tapping to an isochronous sequence and paced tapping to music tasks. Participants were asked to tap on their spacebar to the beat for all production tasks. The un-paced tapping task requires participants to tap at a comfortable rate with no accompanying stimulus for 60s. Participants are asked to try and keep the tapping rate constant throughout the 60s. The paced tapping to an isochronous sequence task asks participants to tap to a metronome for 36s. In the paced tapping to music task, participants are to tap for 36s to four musical excerpts. The measures for the production tasks are inter-tap-interval (ITI), coefficient of variation (CV) and synchronization consistency (vector length). In order to reduce data skewness typical of synchronization data, all vector lengths were submitted to logit transformation prior to analysis [40]. Out of the 148 participants from the full control sample, eight did not complete the task due to technical errors, and 12 failed to follow instructions and were subsequently excluded from the analysis, resulting in a total of 128 participants having results for the BAASTA.

To assess for possible impairments in pitch perception, we used the MBEA [27]. Congenital amusia refers to the limited ability to accurately perceive pitch that is not better explained by hearing loss, neurological damage, or intellectual deficits [41]. Online versions of the scale, contour, and interval tests of the MBEA were employed in the present study to screen participants for congenital amusia, as has been done in previous studies [41]. Participants with amusia were identified as individuals with an average composite score of below 21.36 across all three tests, as outlined in previous work [41]. For descriptive statistics of the screening measures please see Table 1.

## Stimuli

Participants were instructed to listen to the 54 musical stimuli (Fig 1) made to elicit groove taken from Matthews et al. (2019) [8]. Due to technical errors, three stimuli were not played for all participants, which resulted in only 51 included in the final analysis. All stimuli were computer-generated, used piano timbre and were composed using Cubase Pro (v8.0.30). Each

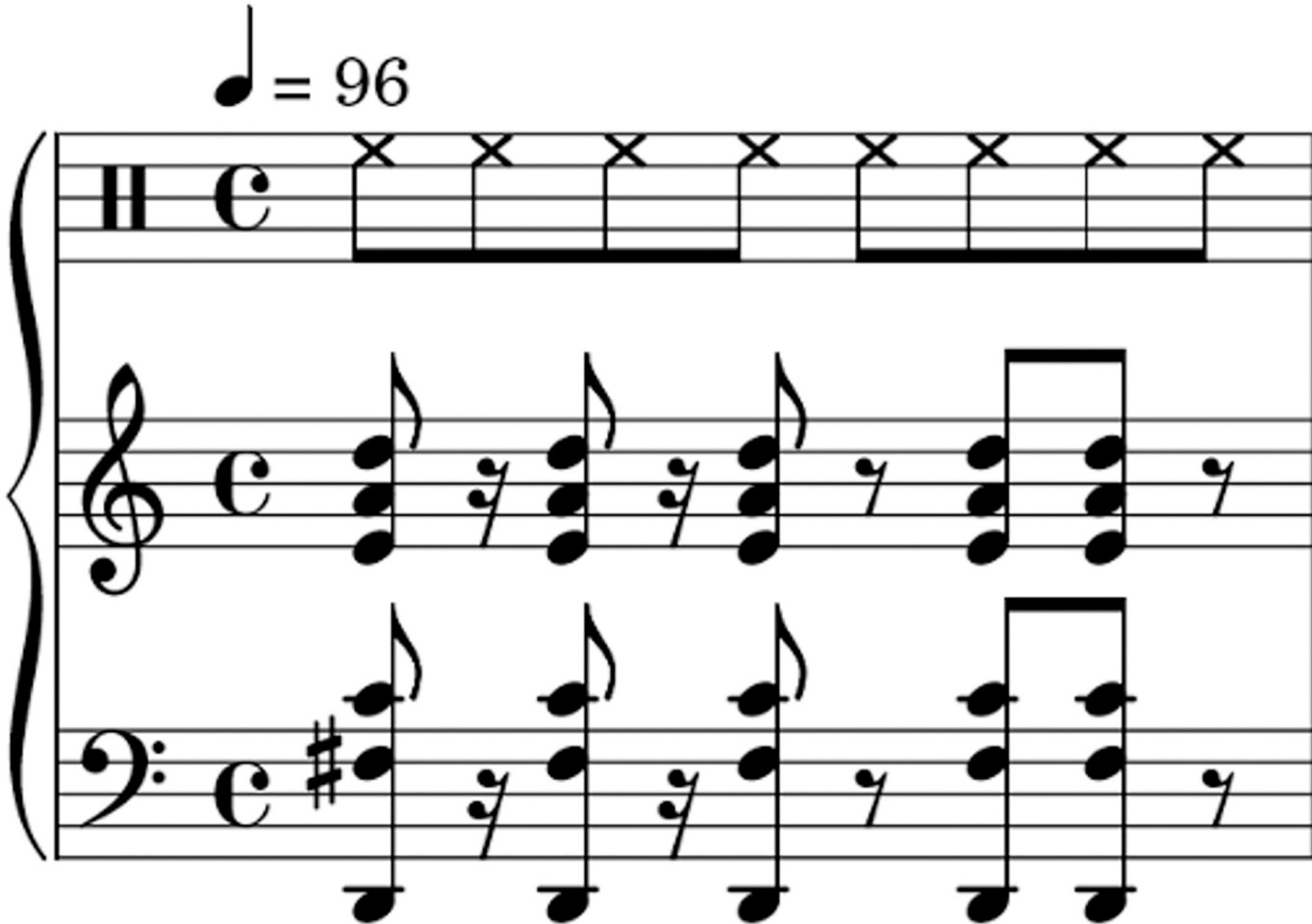

**Fig 1. Example of the musical notation of a stimulus with a medium rhythmic complexity (son clave) and a medium harmonic complexity (four-note (major) chords with extensions).** The upper bar denotes the hi-hat. Figure originated from Matthews et al., 2019. Additional musical notation examples of other stimuli can be found in S1 & S2 Figs.

stimulus lasted for a total of 10 seconds. These stimuli varied in rhythmic and harmonic complexity across three levels: low (17 stimuli for rhythmic complexity; 15 for harmonic complexity), medium (18 stimuli for rhythmic complexity; 18 for harmonic complexity), and high (16 stimuli for rhythmic complexity; 18 for harmonic complexity). Rhythmic complexity was manipulated through the degree of syncopation present in the sequence. Each rhythmic pattern consisted of five onsets. Additionally, each stimulus consisted of four repetitions of each rhythmic pattern. Medium rhythmic complexity stimuli included the son clave, rumba clave and an experimenter-created rhythm. Low rhythmic complexity stimuli were rhythms with the syncopation removed, and high rhythmic complexity had only the first onset fall on the strong beat. Harmonic complexity was influenced by manipulating the chords present in each musical stimulus. All chords were in the key of D. For low harmonic complexity, chords were created using the D major triad and two inversions. Medium-complexity chords were composed of four-note extended chords. Lastly, for high-complexity we added a flat ninth above the root of the chord. These stimuli have been used in various previous studies investigating groove and have been found to be sufficient to elicit the groove response [8, 9, 11]. All musical stimuli used in the study can be found in S1 Dataset.

The order of presentation for the musical stimuli was randomized per participant. Following the presentation of each stimulus, participants were asked to rate on a 5-point Likert scale "How much pleasure do you experience listening to this musical pattern?" and "How much does this musical pattern make you want to move?", with one indicating low levels of derived pleasure/wanting to move and five indicating high levels. The order in which these two questions were presented was randomized and asked sequentially on separate pages. To ensure participants responded to each question appropriately, images were included to symbolize either pleasure or wanting to move (See S3 & S4 Figs).

### Demographics and musical background questionnaire

Participants were asked to complete a short questionnaire assessing their demographics and musical experience. Participants' age and gender were collected through the questionnaire, while their country of origin was collected through Prolific. Data was collected surrounding participants' history playing music, including if they have ever taken lessons for an instrument or singing, if they are currently playing an instrument, the age they started, the age they finished, and the hours a week they play. The same questions were asked of participants regarding their experience with dance.

### Procedure

All measures, excluding the BAASTA subtests, were coded on PsychoPy (v2021.2.3) and hosted on Pavlovia. The BAASTA tasks were hosted on the International Laboratory for Brain, Music, and Sound Research (BRAMS) Online Testing Platform (OTP). To limit burnout, the experiment was split into three separate modules. Module 1 comprised the consent form, the demographics and musical background questionnaire, the BMRQ, and the MBEA. Module 2 contained the WAS, the BDI-II, and the musical rating task. The third and final module consisted of the BAT and the paced tapping to music task. Modules 1 and 2 took approximately 40 minutes to complete, while module 3 took approximately 20 minutes.

### Data analysis

All analyses and graphs were conducted using R (v4.1.2) and RStudio. Correlations were compared using the *cocor* package [42]. Linear mixed-effects models were conducted using the lme4 [43] and *lmerTest* [44] packages. Variance was estimated using restricted maximum likelihood. Degrees of freedom were estimated using the Satterthwaite method [45]. All linear mixed effects included by-participant random intercept. Ratings were averaged across the three versions within each level of complexity. Models were built in a hierarchal fashion, starting with an intercept-only model with no fixed effects and adding variables to the model and comparing if there is an increase in model fit indices through likelihood ratio tests. The base model contained only the by-participant random intercept. Following this, variables were entered in the following order: rhythmic complexity, harmonic complexity, and lastly, any additional variables. Rhythmic and harmonic complexity were added as quadratic predictors for all models. Variables were preselected based on our hypotheses and a to replicate the previous findings (Matthews et al., 2019). As such, we did not check for other options or combinations. Due to the small size of the musical anhedonia sample and restrictions from analysis packages, random slopes and a random intercept for stimuli were not able to be included in the models. Contrast analysis was employed to investigate the effect between the levels of rhythmic and harmonic complexity using the R package *emmeans* [46]. Mediation analyses were conducted using the *mediation* package [47]. Mediation analyses were conducted using

1000 simulations. All models were assessed to ensure they met appropriate statistical assumptions. Significance was determined at the cut-off of p-values less than 0.05.

## Results

### Full control sample

For visualization of the data, please see Fig 2. The full control sample consisted of the remaining 148 participants after excluding those with musical anhedonia. For pleasure ratings, the addition of rhythmic complexity ($\chi^2(2) = 260.34$, $p < 0.001$), harmonic complexity ($\chi^2(2) = 166.41$, $p < 0.001$), and the interaction between rhythmic and harmonic complexity ($\chi^2(2) = 45.58$, $p < 0.001$) as predictors to the model significantly increased the model fit indices (see S2 Table). Results indicated a significant interaction between rhythmic and harmonic complexity on pleasure ratings ($b(1048) = 0.092$, $p < 0.001$, 95% CI [0.043, 0.141]). Contrast analyses indicated that both rhythmic ($b(894) = -1.53$, $p < 0.001$, 95% CI [-1.63, -1.43]) and harmonic complexity ($b(894) = -0.14$, $p = 0.005$, 95% CI [-0.237, -0.043]) contained significant quadratic relationships. Contrast analyses indicated that medium rhythmic complexity resulted in significantly higher ratings of pleasure compared to low rhythmic complexity ($b(575) = -0.382$, $p < 0.001$, 95% CI [-0.448, -0.316]), while low complexity resulted in higher ratings of pleasure compared to high complexity ($b(149) = 0.56$, $p < 0.001$, 95% CI [0.479, 0.64]). While the quadratic relationship for harmonic complexity is significant, the results from the contrast analyses support more of a linear relationship between the levels of harmonic complexity. Low harmonic complexity resulted in significantly higher ratings of pleasure compared to medium complexity ($b(666) = 0.21$, $p < 0.001$, 95% CI [0.147, 0.273]), and medium harmonic complexity resulted in higher ratings compared to high complexity ($b(666) = 0.35$, $p < 0.001$, 95% CI [0.287, 0.413]).

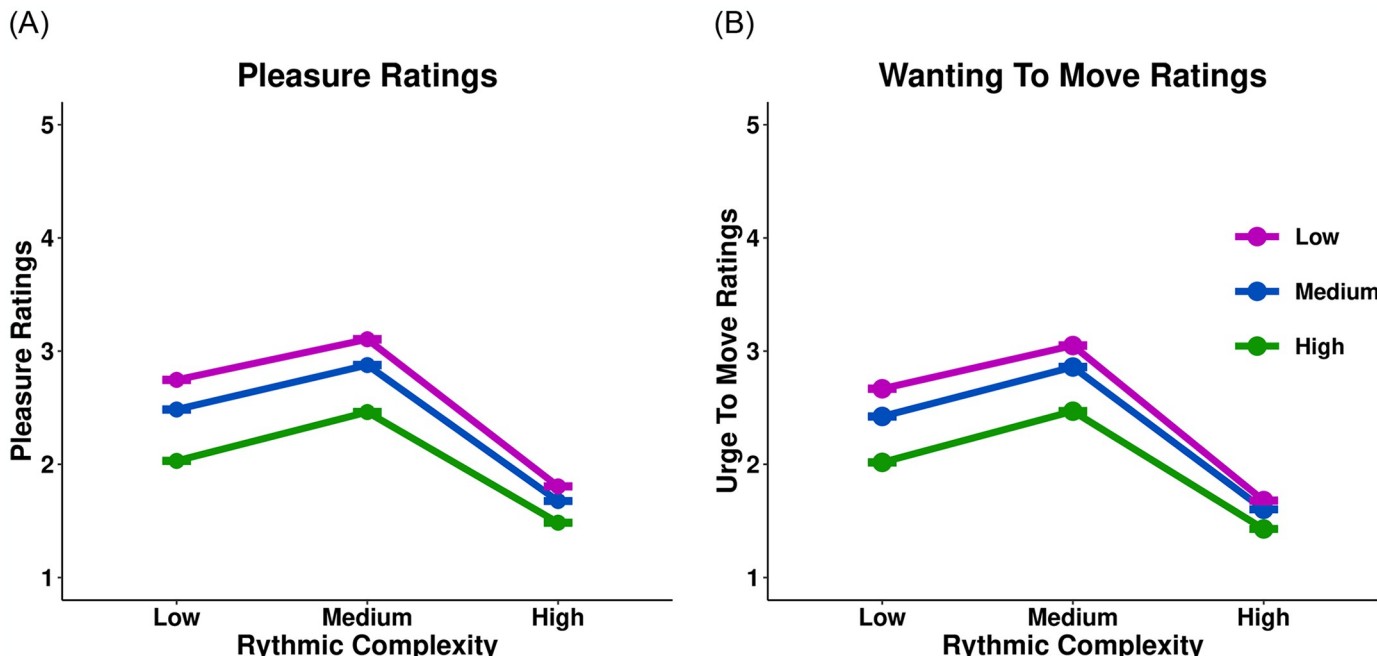

**Fig 2. The effect of rhythmic and harmonic complexity on groove response for the full control sample.** The y-axis represents pleasure or wanting to move ratings on a 5-point Likert scale. The x-axis represents the rhythmic complexity of the musical samples. Colours indicates the level of harmonic complexity of the musical sample. Dots represent means derived from raw values.

For wanting to move ratings, adding rhythmic complexity ($\chi^2(2)$ = 272.72, $p < 0.001$), harmonic complexity ($\chi^2(2)$ = 120.82, $p < 0.001$) and the interaction between rhythmic and harmonic complexity ($\chi^2(2)$ = 11.74, $p < 0.001$) as predictors to the model significantly increased the model fit indices (S2 Table). Results indicated a significant interaction between rhythmic and harmonic complexity on wanting to move ratings ($b(1197)$ = 0.092, $p < 0.001$, 95% CI [0.039, 0.145]). Similar to pleasure ratings, the contrast analyses indicated a significant quadratic trend for both rhythmic ($b(1195)$ = -0.896, $p = 0.01$, 95% CI [-0.989, -0.803]) and harmonic complexity ($b(1328)$ = -0.144, $p = 0.01$, 95% CI [-0.253, -0.035]). Wanting to move ratings follow a similar trend to pleasure ratings regarding rhythmic complexity, as medium complexity resulted in significantly higher ratings compared to low complexity ($b(1195)$ = -0.122, $p < 0.001$ 95% CI [-0.176, -0.068]), and low resulted in significantly higher ratings compared to high complexity ($b(1195)$ = 0.652, $p < 0.001$, 95% CI [0.598, 0.706]). While the quadratic trend was significant for harmonic complexity, like with pleasure ratings, the contrast analyses demonstrate more of a linear relationship. Low harmonic complexity resulted in significantly higher wanting to move ratings compared to medium complexity ($b(1195)$ = 0.177, $p < 0.001$, 95% CI [0.114, 0.24]), while medium complexity resulted in significantly higher ratings compared to high complexity ($b(1195)$ = 0.774, $p < 0.001$, 95% CI [0.72, 0.828]).

It is counterintuitive to think that harmonic complexity affects wanting to move ratings independently. Therefore, it would make sense that harmonic complexity affects wanting to move ratings via its effect on pleasure ratings. This concept is further supported by previous research using the same stimuli, which demonstrated that pleasure ratings mediate the effect of harmonic complexity on wanting to move ratings [8]. We conducted the same analyses in the current sample. For rhythmic complexity, when pleasure ratings are added into the model, the effect on wanting to move ratings significantly decreased. An indirect effect (*IE*) emerged (*IE* = -0.33, $p < 0.001$, 95% CI [-0.368, -0.29]), but the average direct effect (*DE*) remained significant (*DE* = -0.03, $p < 0.001$, 95% CI [-0.049, -0.01]). While the direct effect remained statistically significant, the size of the effect diminished substantially.

When pleasure was introduced into the model as a predictor, the effect of harmonic complexity on wanting to move ratings, an indirect effect emerged (*IE* = -0.218, $p < 0.001$, 95% CI [-0.218, -0.18], and the direct effect became non-significant (*DE* = 0.017, $p = 0.064$, 95% CI [-0.002, 0.03]. These results replicate the findings of Matthews et al., 2019, [8] supporting the idea that harmonic complexity influences wanting to move ratings primarily through its effect on pleasure ratings.

## Musical anhedonia and match sample comparisons

For visualization of the data, please see Fig 3. Consistent with the results for the control sample, adding rhythmic ($\chi^2(2)$ = 19.13, $p < 0.001$) and harmonic complexity ($\chi^2(2)$ = 19.65, $p < 0.001$) as predictors to the models predicting pleasure ratings within the musical anhedonia sample significantly increases the model fit indices. Unlike the full sample, the addition of the interaction effect between rhythmic and harmonic complexity did not result in a significant improvement to the model fit ($\chi^2(2)$ = 2.16, $p = 0.142$; See S2 Table). The same is seen for the model predicting wanting to move ratings (Rhythmic complexity: $\chi^2(2)$ = 20.82, $p < 0.001$, Harmonic complexity: $\chi^2(2)$ = 17.22, $p < 0.001$, Rhythmic/Harmonic Complexity Interaction: $\chi^2(2)$ = 2.9, $p = 0.085$).

The same contrast analyses conducted for the full control sample were conducted for the musical anhedonia sample. Rhythmic complexity showed a significant quadratic relationship with both pleasure ($b(132)$ = -1.630, $p < 0.001$, 95% CI [-1.97, -1.29]) and wanting to move ($b(132)$ = -1.57, $p < 0.001$, 95% CI [-1.88, -1.26]) ratings. Unlike in the full sample, in the musical

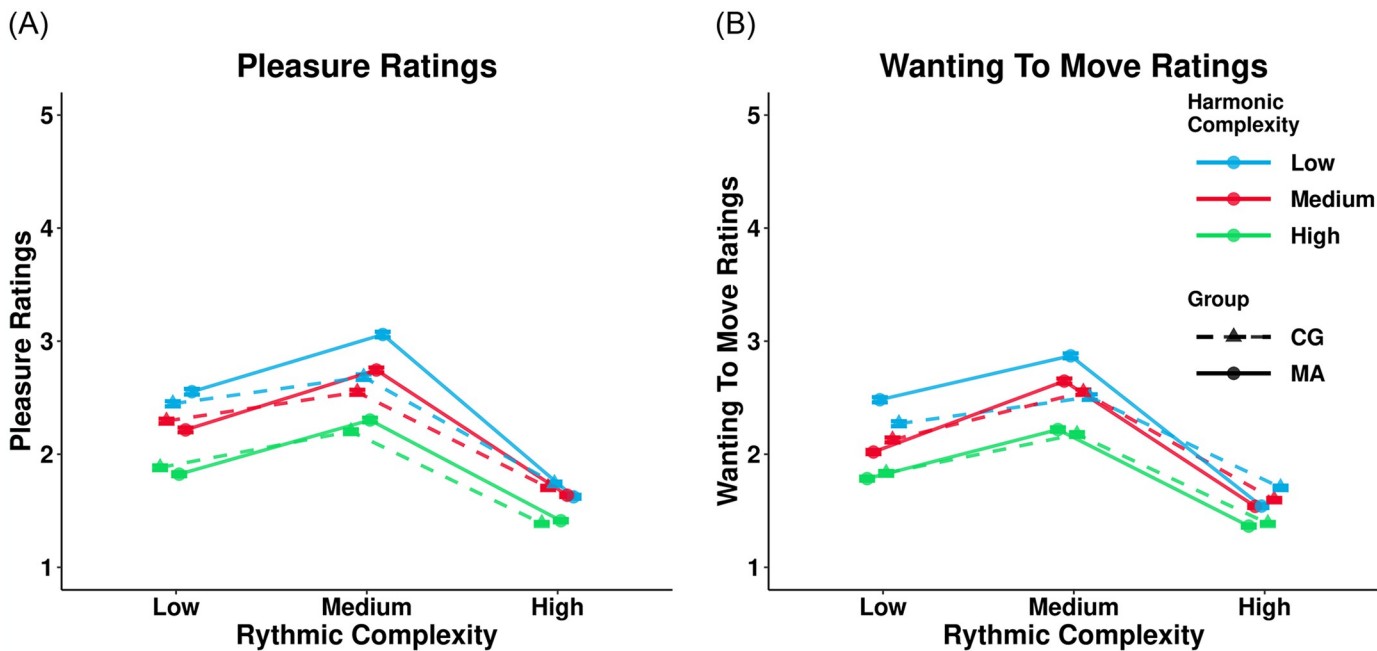

**Fig 3. The effect of rhythmic and harmonic complexity on groove response for the musical anhedonia.** The musical anhedonia sample values were adjusted by 0.2 on the x and y- axes for graphical representation. The y-axis represents pleasure or wanting to move ratings on a 5-point Likert scale. The x-axis represents the metric complexity of the musical samples. Colours indicates the level of harmonic complexity of the musical sample. Shapes and line type indicates if it is either the musical anhedonia sample, or the matched control sample. Dots represent means derived from raw values.

anhedonia sample, harmony did not show a significant quadratic relationship with either wanting to move ($b$(132) = -0.043, $p$ = 0.79, 95% CI [-0.358, 0.272]) or pleasure ratings ($b$(132) = -0.122, $p$ = 0.482, 95% CI [-0.461, 0.217]). Besides this, the pattern of results was identical to those seen in the analysis of the full sample (see Table 3). The matched control group followed the same pattern of results seen for the musical anhedonia (see Tables 2 & 3).

To compare the musical anhedonia and matched control group sample, an effect of group and its interaction were added to both the pleasure and wanting to move models. Results from the models indicated that there was no significant difference between the musical anhedonia and matched control group or between the interactive effects of rhythmic or harmonic

**Table 2. Linear mixed effect model results for the musical anhedonia and matched control sample.**

| | \multicolumn{5}{c}{*Pleasure Ratings*} | \multicolumn{5}{c}{*Wanting To Move Ratings*} |
| | *df* | *b* | *p* | *Lower Bound 95% CI* | *Upper Bound 95% CI* | *df* | *b* | *p* | *Lower Bound 95% CI* | *Upper Bound 95% CI* |
|---|---|---|---|---|---|---|---|---|---|---|
| RC | 266 | -0.645 | 0.062 | -1.32 | 0.029 | 266 | -0.749 | 0.027* | -1.41 | -0.088 |
| HC | 266 | -0.687 | 0.047* | -1.36 | -0.013 | 266 | -0.753 | 0.026* | -1.41 | -0.092 |
| GRP | 261 | -0.411 | 0.437 | -1.45 | 0.62 | 263 | -0.608 | 0.24 | -1.62 | 0.404 |
| RC ~ HC | 266 | 0.163 | 0.307 | -0.149 | 0.475 | 266 | 0.204 | 0.194 | -0.102 | 0.509 |
| RC ~ GRP | 266 | 0.106 | 0.625 | -0.32 | 0.532 | 266 | 0.194 | 0.365 | -0.224 | 0.612 |
| HC ~ GRP | 266 | 0.166 | 0.45 | -0.26 | 0.592 | 266 | 0.244 | 0.254 | -0.174 | 0.662 |
| RC ~ HC ~ GRP | 266 | -0.049 | 0.63 | -0.246 | 0.149 | 266 | -0.077 | 0.434 | -0.271 | 0.116 |

RC = Rhythmic Complexity, HC = Harmonic Complexity, GRP = Group, ~ signifies an interaction, *df* = Degrees of freedom, CI = Confidence Intervals, * indicates *p*-value < 0.05.

**Table 3. Contrast analyses for the musical anhedonia and matched control sample.**

| | | Pleasure Ratings | | | | | Wanting To Move Ratings | | | |
|---|---|---|---|---|---|---|---|---|---|---|
| | df | b | p | Lower Bound 95% CI | Upper Bound 95% CI | df | b | p | Lower Bound 95% CI | Upper Bound 95% CI |
| MA: LR—MR | 132 | -0.505 | < 0.001* | -0.701 | -0.309 | 132 | -0.483 | < 0.001* | -0.665 | -0.301 |
| MA: MR—HR | 132 | 1.126 | < 0.001* | 0.93 | 1.322 | 132 | 1.09 | < 0.001* | 0.908 | 1.272 |
| MA: LH—MH | 132 | 0.232 | 0.0564 | 0.036 | 0.428 | 132 | 0.236 | 0.032* | 0.054 | 0.418 |
| MA: MH—HH | 132 | 0.354 | 0.002* | 0.158 | 0.55 | 132 | 0.279 | 0.009* | 0.097 | 0.461 |
| CG: LR—MR | 132 | -0.271 | < 0.001* | -0.387 | -0.155 | 140 | -0.333 | 0.001* | -0.508 | -0.158 |
| CG: MR—HR | 132 | 0.871 | < 0.001* | 0.757 | 0.985 | 140 | 0.862 | < 0.001* | 0.687 | 1.037 |
| CG: LH—MH | 132 | 0.088 | 0.303 | -0.029 | 0.204 | 140 | 0.049 | 0.848 | -0.126 | 0.224 |
| CG: MH—HH | 132 | 0.353 | < 0.001* | 0.24 | 0.466 | 140 | 0.288 | 0.005* | 0.113 | 0.463 |

MA = Musical Anhedonia Sample, CG = Matched Control Sample, LR = Low Rhythmic Complexity, MR = Medium Rhythmic Complexity, HR = High Rhythmic Complexity, LH = Low Harmonic Complexity, MH = Medium Harmonic Complexity, HH = High Harmonic Complexity, df = Degrees of freedom, CI = Confidence Intervals,

* indicates p-value < 0.05.

complexity (see Table 2). Additionally, the correlation between pleasure and wanting to move ratings was similar between the musical anhedonia ($r = 0.939$) and matched control group ($r = 0.914$), and not significantly different ($z = 0.472$, $p = 0,637$).

## Mediation analysis—Musical anhedonia and matched control sample

We expected that individuals with musical anhedonia would experience lower perceived pleasure from music. Therefore, we hypothesized that the relationship between rhythmic and harmonic complexity and pleasure responses was maintained through the relationship via wanting to move ratings. A mediation analysis was conducted to determine if wanting to move ratings mediated the effect of rhythmic and harmonic complexity on pleasure ratings. When wanting to move was added to the model, the significant direct effects of both harmonic ($DE = -0.044$, $p = 0.12$, 95% CI [-0.097, 0.01]) and rhythmic ($DE = -0.014$, $p = 0.62$, 95% CI [-0.069, 0.04]) complexity on pleasure ratings became non-significant, indicating that wanting to move ratings completely mediated these effects (Harmonic complexity: $IE = -0.287$, $p < 0.001$, 95% CI [-0.421, -0.17]; Rhythmic complexity: $IE = -0.303$, $p < 0.001$, 95% CI [-0.432, -0.17]). Contrast analyses indicated that when wanting to move was added to the model, all of the contrast analyses became non-significant (see Table 4). In order to investigate if this same effect was present in the matched controls, the same mediation analysis was run on the matched control sample. Unlike with the musical anhedonia sample, when wanting to move ratings were added to the model, the significant direct effects of both rhythmic ($DE = -0.073$, $p = 0.008$, CI [-0.124, -0.02]) and harmonic complexity ($DE = -0.079$, $p = 0.002$, CI [-0.126, -0.03]) were significantly reduced, and an indirect effect emerged for both (Rhythm: $IE = -0.257$, $p < 0.001$, CI [-0.349, -0.17]; Harmony: $IE = -0.192$, $p < 0.001$, CI [-0.282, -0.11]). Contrast analyses indicated that when wanting to move ratings were added to the model, some of the contrast analyses, but not all became non-significant, supporting partial mediation (see Table 4).

## Discussion

The present study set out to investigate if pleasure and the urge to move can be separated from one another in the sensation of groove. We did this by comparing ratings of wanting to move and pleasure in a large sample of controls and in people with specific musical anhedonia. We

**Table 4. Contrast analyses for the mediation analysis for the musical anhedonia and matched control samples.**

|  | Musical Anhedonia Group | | | | | Matched Control Group | | | | |
|---|---|---|---|---|---|---|---|---|---|---|
|  | df | b | p | Lower Bound 95% CI | Upper Bound 95% CI | df | b | p | Lower Bound 95% CI | Upper Bound 95% CI |
| LR—MR | 135 | -0.049 | 0.668 | -0.161 | 0.631 | 130 | 0.006 | 0.993 | -0.092 | 0.104 |
| LR-HR | 138 | 0.049 | 0.69 | -0.068 | 0.166 | 134 | 0.165 | 0.006* | 0.611 | 0.269 |
| MR—HR | 143 | 0.098 | 0.367 | -0.044 | 0.24 | 140 | 0.16 | 0.026* | 0.041 | 0.279 |
| LH—MH | 129 | 0.01 | 0.983 | -0.096 | 0.125 | 127 | 0.055 | 0.484 | -0.039 | 0.099 |
| LH-HH | 136 | 0.101 | 0.192 | -0.012 | 0.214 | 131 | 0.171 | 0.002* | 0.073 | 0.269 |
| MH—HH | 130 | 0.091 | 0.218 | -0.015 | 0.198 | 130 | 0.116 | 0.053 | 0.019 | 0.213 |

LR = Low Rhythmic Complexity, MR = Medium Rhythmic Complexity, HR = High Rhythmic Complexity, LH = Low Harmonic Complexity, MH = Medium Harmonic Complexity, HH = High Harmonic Complexity, df = Degrees of freedom, CI = Confidence Intervals,

* indicates p-value < 0.05.

hypothesized that if wanting to move and pleasure are separable, individuals with specific musical anhedonia would show reduced ratings of pleasure but preserved ratings of wanting to move. Instead, those with musical anhedonia showed the same inverted U-shaped response to musical stimuli that elicit groove for both pleasure and wanting to move as controls. We then used mediation analysis to show that the pleasure response in those with musical anhedonia was fully mediated by wanting to move ratings, whereas the pleasure response in controls showed only partial mediation. This pattern of results indicates that for those with musical anhedonia the urge to move preserves their pleasure response. More broadly, these findings suggest that the desire to move may itself generate pleasure, contributing directly to the experience of groove.

For both ratings of wanting to move and pleasure we found an inverted U-shaped pattern of responses for rhythmic complexity across all our samples such that medium levels of complexity produced the highest ratings. This is consistent with previous studies using the same stimuli [8] and with other studies using either real musical clips or drum breaks [7, 15, 20]. Inconsistent with results from Matthews et al., 2019 [8], our contrast analyses indicated that low harmonic complexity induced significantly higher ratings of pleasure and urge to move compared to medium harmonic complexity. These findings are, however, consistent with those found by Stupacher et al., 2022 [9]. The mediation analyses conducted for the full sample also replicated the findings of Matthews et al., 2019 [8]. Pleasure ratings acted as a partial mediator for the effect of rhythmic complexity on urge to move ratings. The effect of harmonic complexity on the urge to move ratings was also fully mediated by pleasure ratings. Both results support the concept that pleasure and urge to move ratings are partially separable from one another.

Surprisingly, we found no difference in pleasure or urge to move ratings between the musical anhedonia sample and the matched control sample, either in terms of overall ratings or in the inverted-U-shaped pattern across levels of rhythmic complexity. This indicates that the overall blunted pleasure response to music in those with musical anhedonia does not affect their ability to derive pleasure from stimuli that elicit groove. We hypothesized that the preserved quadratic relationship may be based on pleasure derived from the urge to move in these participants. To test this idea, we conducted a mediation analysis on the musical anhedonia sample and demonstrated that the wanting to move ratings completely mediated the effect of both harmonic and rhythmic complexity on pleasure ratings. These results support the idea that the groove response may be maintained in those with musical anhedonia through the urge to move ratings. This is partially consistent with recent findings showing that individuals with musical anhedonia had overall reduced pleasure ratings but preserved urge to move

ratings for drum-breaks derived from real music [48]. The dampening of overall pleasure responses for those with musical anhedonia in this study may be due to two possible reasons. For one, the more varied stimuli may have provided additional information that influenced participants' pleasure ratings. Second, the more lenient inclusion criteria for the musical anhedonia group used in this study may have allowed for individuals with musical perceptual deficits, depression, or reduced general reward sensitivity to be included, which could have impacted the pleasure ratings.

More broadly, these results imply that the urge to move is a central component of groove that differentiates it from other modalities of enjoying music. A potential mechanism could be that part of the pleasure derived from groove originates from the urge to move through learning and refinement of our motor models [12, 17–19]. As previously discussed, predictive coding conceptualizes the urge to move as the attempt by the brain to minimize the prediction error between our top-down model and the bottom-up sensory information [12, 17–19]. This is supported by previous work that has demonstrated that higher perceived accuracy on tapping tasks was associated with greater pleasure ratings [49]. Our findings also align with the model proposed by Senn and colleagues in 2019 [14]. In particular, these results support their hedonic feedback loop. If the pleasure is being driven by the urge to move, then this would make sense why individuals classified as having musical anhedonia would still experience pleasure from groove.

This idea of the urge to move being the producer of the pleasure response in groove makes sense to explain why we may see the observed pattern of responses in those with musical anhedonia. A previous brain imaging study found that individuals with musical anhedonia had decreased functional connectivity between the right auditory cortex and the ventral striatum [23]. Areas of the ventral striatum, such as the NAcc, are related to the processing of reward stimuli [11, 50–52]. Given this disruption in functional connectivity between the auditory cortex and ventral striatum, one would expect to see lower ratings for pleasure from music. The sensation of groove could rely more on temporal predictive process supported by motor and premotor areas. Work has demonstrated that the anticipation of beat and beat perception relies on the motor system including the dorsal striatum [53–56]. This is consistent with the findings of Matthews et al. (2020) [11], which that both the dorsal and ventral striatum were active in response to high groove stimuli and that activity in the dorsal striatum was primarily related to ratings of urge to move.

## Conclusion

The present study found that individuals with musical anhedonia demonstrate a preserved pleasure response for groove stimuli that is potentially mediated through the urge to move. The urge to move has been linked to dorsal striatal reward circuits, while pleasure is more associated with ventral striatal circuits [11]. Thus, we hypothesize that the groove response in anhedonia is preserved through dorsal striatal connections with motor and auditory regions. More generally, our results suggest that the sensation of groove may be different from other types of musical pleasure in that the urge to move may be a primary source of the pleasurable experience. This thinking aligns with definitions of the groove sensation that emphasize the role of the urge to move [5, 6]. Future work examining possible intrinsic reward value of movement could shed light on these findings.

## Supporting information

**S1 Dataset. Musical stimuli used in the groove rating task.**
(ZIP)

**S1 Fig. Additional example of music notation of stimuli high rhythmic and harmonic complexity.**
(TIF)

**S2 Fig. Additional example of music notation of stimuli low rhythmic and harmonic complexity.**
(TIF)

**S3 Fig. Image included alongside wanting to move question.**
(TIF)

**S4 Fig. Image included alongside pleasure question.**
(TIF)

**S1 Table. Demographics.** M = mean, SD = standard deviation.
(DOCX)

**S2 Table. Likelihood ratio test results.** $X^2$ = Chi-Square, RC = Rhythmic Complexity, HC = Harmonic Complexity, ~ indicates an interaction, Harmonic complexity was compared to the model with rhythmic complexity as a predictor.
(PDF)

## Acknowledgments

The authors would like to thank Maria Psomas, Marin Hoh, and Nigel Ward for their help with participant recruitment. They would also like to thank Alexander Albury, Connor Spiech and Maria Witek for their practical help and advice.

## Author Contributions

**Conceptualization:** Isaac D. Romkey, Tomas Matthews, Virginia B. Penhune.

**Data curation:** Isaac D. Romkey, Tomas Matthews, Nicholas Foster.

**Formal analysis:** Isaac D. Romkey, Nicholas Foster.

**Funding acquisition:** Virginia B. Penhune.

**Investigation:** Isaac D. Romkey.

**Methodology:** Isaac D. Romkey, Tomas Matthews, Nicholas Foster, Virginia B. Penhune.

**Project administration:** Isaac D. Romkey.

**Resources:** Simone Dalla Bella, Virginia B. Penhune.

**Software:** Isaac D. Romkey.

**Supervision:** Isaac D. Romkey, Virginia B. Penhune.

**Validation:** Isaac D. Romkey.

**Visualization:** Isaac D. Romkey.

**Writing – original draft:** Isaac D. Romkey.

**Writing – review & editing:** Isaac D. Romkey, Tomas Matthews, Nicholas Foster, Simone Dalla Bella, Virginia B. Penhune.

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
