## [Decision Letter · Decision Letter 0]

17 Jun 2024

PONE-D-24-06715The Pleasurable Urge to Move to Music is Unchanged in Musical AnhedoniaPLOS ONE

Dear Dr. Romkey,

Thank you for submitting your manuscript to PLOS ONE. After careful consideration, we feel that it has merit but does not fully meet PLOS ONE’s publication criteria as it currently stands. Therefore, we invite you to submit a revised version of the manuscript that addresses the points raised during the review process.

We look forward to receiving your revised manuscript.

Kind regards,

Hira Rafi

Academic Editor

PLOS ONE

Journal Requirements:

 [This research was supported by a grant to VP from the Natural Sciences and

Engineering Research Council of Canada

(NSERC 2021-04026). IR was funded by the Canadian Graduate Scholarship -

Masters by NSERC and by the Fonds de Rechereche du Québec - Nature et

Technologies (FRQ - NT). SDB and NF received funding from a Discovery Grant

(RGPIN-2019-05453) from NSERC, and by the Canada Research Chair program (CRC

in music auditory-motor skill learning and new technologies).].  

[SDB is on the board of the BeatHealth company dedicated to the design and commercialization of technological tools for assessing rhythm capacities such as

BAASTA and implementing rhythm-based interventions.]. 

Reviewers' comments:

Reviewer's Responses to Questions

**Comments to the Author**

1. Is the manuscript technically sound, and do the data support the conclusions?

Reviewer #1: Partly

Reviewer #2: Partly

Reviewer #3: Partly

2. Has the statistical analysis been performed appropriately and rigorously? 

Reviewer #1: N/A

Reviewer #2: I Don't Know

Reviewer #3: No

3. Have the authors made all data underlying the findings in their manuscript fully available?

Reviewer #1: Yes

Reviewer #2: Yes

Reviewer #3: Yes

4. Is the manuscript presented in an intelligible fashion and written in standard English?

Reviewer #1: Yes

Reviewer #2: Yes

Reviewer #3: Yes

5. Review Comments to the Author

Reviewer #1: I have mixed feelings about this article. On one hand, it follows a clever idea to separate urge to move and pleasure in groove, and employs a high qualitative research design to do so. On the other hand, the literature review, theoretical underpinning, and discussion are somewhat lacking and there are some issues with the analysis.

Major issues:

1. I see some issues with how the model and variable selection were performed; it would be worthwhile to do some more testing and update the models if necessary.

a) Variable selection: you conducted forward addition, but as far as I know, backwards elimination or bidirectional addition/elimination is more common. The latter is in any case the more thorough method and should be preferred.

b) I don’t see a test for the model structure. You state that your baseline model had a by-participant intercept, which is reasonable for a start - although you do not provide your theoretical reasoning for it. However, it seems you did not check for other options. It is counterintuitive to expect that the stimuli are sufficiently described by HC and RC, and that a by-stimulus intercept would thus be unnecessary. Similarly, a slope for RC can be expected: why would every participant react with the same strength to the manipulations of RC? I did a quick test for some of the possible configurations with your data, and it seems that a model with (Rhythm | ID) + (1 | MusicName) would be the most efficient model structure (but I might have made a mistake, and I did not check for the groups independently!). I guess this model will run into convergence issues with your MA subset due to the low amount of data. I am also aware that the mediation package is not able to work correctly with slopes. I have not calculated all your models, and can’t say whether different models would change some of your results. Yet, I think it is necessary to check for additional model structures, at least for the by-stimulus intercept. If you choose to ignore better fits with slopes, you can probably find an argumentation for that, as a) it is not uncommon to skip the respective checks, i.e., provide an argumentation that other papers did not use it and why you think this is the correct way to handle it or b) mediation analysis is central to your study and, as far as I know, this gets a bit messy with random slopes, i.e., provide an argumentation based on your intended method and why you think results are not changed by dropping down to your current model.

2. The paper is at times very repetitive. In some cases, you are repeating half sentences that you just brought up one or two sentences above. In others, full sentences are almost duplicates. It would enhance the readability of your paper if you could cut down on repetitions.

3. Your literature review is one-sided (even more emphasized by frequent mentions of the own lab) and misses some studies that examined the relationship of pleasure and urge to move that should be taken into account. For example, Senn et al. (2020) tried to separate urge to move and pleasure through a specific set of real music stimuli but failed, probably due to methodological problems. In Senn et al. (2023), the correlation between pleasure and urge to move was rather low and pleasure hardly predicted any variance in urge to move ratings, again using stimuli selected to show independence. And probably of most interest to your study is the psychological model of groove (Senn et al., 2019; later adapted) that postulates a so far untested for hedonistic feedback loop between urge to move and pleasure in groove. Their reasoning and argumentation could be quite useful for your introduction and discussion, especially as your study seems to confirm this loop/the half that has not been tested before. In my opinion, your study would benefit greatly from considering these results – and not including some of these leaves the impression that of a hastily literature review process.

a) You are not sufficiently introducing the idea of a non-separable urge to move and pleasure. On what basis would you think that they would be either the same thing or fuse to a single entity? In case it is because of the high correlation, then an r = 0.8 means that pleasure still cannot explain 36% of the variance in urge to move (and this amount likely increases if you add a random intercept). Hence, I don’t think this is sufficient as an explanation. And besides this strong correlation, Senn et al. (2020) and Düvel et al. (2021) found that the best fitting CFA on their 6-item questionnaire on groove has in fact two constructs (pleasure and urge to move), and not one. This is quite a good argument that they are separable already.

b) Pleasure is not sufficiently introduced in the introduction nor discussed later. You provide some hints that you subscribe to the idea that multiple pleasures exist or at least pleasure derived from multiple sources. Yet, as this is quite important for your main result and conclusion, this is underdeveloped and lacks proper sources about the nature of pleasure. I’m not aware of a published study that investigates the nature of pleasure in groove in particular; there are some things in Danielsen (2006), Roholt (2014), Janata et al. (2012) and Bechtold et al. (2023) but the last two seemingly preferred the term “positive affect” and I’m not sure if it fits completely with your concept of pleasure. However, there are countless texts on pleasure from many disciplines and your paper would benefit from being clearer on how you actually understand one of the examined core concepts.

c) Musical Anhedonia is also not properly introduced. As a reader, I’m sufficiently informed how you operationalized it in the study, but what it actually is, how it comes about, what body or brain parts it affects, what consequences it has for affected people, etc. is not discussed at all.

4. The method section is very long and, in my opinion, not properly organized. You start with a pseudo-procedure, then go to participants which features many things that have not been introduced before. Then you have a section about rating groove that mostly discusses the stimuli. Afterwards, there is a very long section about measures, followed by a very short section about measures. Then, you have a proper procedure and a data analysis section.

My suggestion to make this more streamlined: drop the initial procedure, and start with a section about all measures, then stimuli, then a proper procedure and last the analysis. And regarding the measures, I think you are giving way too many details that are not particularly interesting to the reader. Their main purpose seems to be to check whether a participant has MA and are consequently not discussed much in the results or discussions. I would suggest to significantly cut down on this (e.g., no need to report the exact procedure, stimuli, nor analysis for these if they are established tests) and put the rest in a supplementary file.

Minor issues:

Title:

I’m not a native speaker, but to me the title sounds incomplete. Unchanged from what? And is “in Musical Anhedonia” correct? Shouldn’t it be “in People with Musical Anhedonia” or something similar.

Abstract:

The abstract is too long and detailed. It should serve as a quick overview over the study and would benefit from being shorter and the main takeaways presented earlier.

o L38: I do not think that N=148 qualifies as large and you do not lose anything by cutting this claim. Especially, N = 17 is not large.

o L38/L44: you are describing your stimuli as rhythmic but then later they apparently have harmonic complexity. It is clear in the paper but incomprehensible in the abstract.

o L47: “response” is very technical, how about “sensation” or “experience”?

Introduction:

- L51: “often generates a…” is too vague.

- L64: please provide the citations after each structural component.

- L71: predictive coding and groove: this theory is en vogue and it might have some importance to your study and thus warrant inclusion. However, the way you summed it up, it does not provide a useful explanation for how groove comes about. If I take just your information, it sounds like I cannot ever experience groove with music where my prediction error is very low (e.g., because I know it very well). This is counterintuitive (and against results regarding familiarity) and the explanation should be corrected to avoid such misunderstandings.

- L78ff: the studies you list do not show a great variety of stimuli, and I don’t think any qualify as natural excerpts, as they are all synthesized. Also, they are all either drum beats or single rhythm stimuli (as there is no syncopation index for polyphonic music other than drums as far as I am aware). Some studies, e.g., Senn et al. (2018) found a different effect of syncopation, Senn et al. (preprint) found none. And while preprints are to be taken with a grain of salt, the latter includes a reanalysis of the Witek et al. (2014) data that showed that the inverted U is not to be found in the more ecologically vallid stimuli, but only when experimenter composed ones are included. Hence, I think your claim about variety and naturalism is problematic.

- L83ff: this section could do with better transitions. It starts with an almost exact repetition and a re-introduction of the topic, but then goes on to present two studies in more detail. Yet, why these two studies are of particular importance is not clear until the reader has read the full paragraph on them. This makes for strange changes of subject on a first read. It would be beneficial to mark the purpose of the section (no reintroduction needed) and why each study is important for the present study’s hypotheses and methods.

- L110: you keep talking about the stimuli, but the reader does not know anything about them yet, besides that they have chords that mark the rhythm. If the stimuli are of such importance here already, please introduce them properly.

- L111: here would be a good place to expand on MA

- L115: here would be a good place to introduce your theory on pleasures/pleasure derived from multiple sources.

Methods:

- L133: “musical history” is misleading, “musical background” instead?

- L161: misleading headline, this paragraph (until L183) should be titled “stimuli.”

- L188: it is unclear whether the two questions were asked on the same page, on following pages, or in separate blocks of the experiment (which is potentially important for the correlation).

- L188f: what images? Can these be included in the supplementary files?

- L205: a duration of 20 “quarter notes”: “beats” might be more appropriate?

- L215: here is info about duration, but you don’t give this for all tasks (redundant if you decide to cut).

- L304ff: please cite the packages that you use properly.

- L305: do you mean “Satterthwaite”?

- L312: “that analysis”: it is unclear what is meant by that.

- L311: it seems that you added RC and HC always as quadratic. Please state so if this is true.

Results:

- L333: RC > HC: please provide statistical proof.

- L374: DE remained significant, but also small/tiny. You have not defined boundaries for interpretation in your text, but maybe it would be good idea to add whether you think it is a relevant effect or neglectable.

- L381: here, you have much, much smaller chi2 compared to the controls. Is this subjectable to the small amount of data or would you have another interpretation (e.g., smaller effect)? In general, I think N = 17 is rather small for a model with random by-participant intercept. Did you perform a power analysis beforehand to check whether your theorized for effect size can be found with this number of participants?

- L398: please test whether the correlations are significantly different and include the results as proof.

- L405: what is metric complexity in this study?

- Table 2-4: please make it clearer where one model ends and the other starts, e.g., with a vertical line.

- L415: you are jumping between presenting results for MA and comparing to the controls. You gave a lot of space to present controls beforehand. I suggest doing the same for the MA group and then turn to the comparison. That way, it is easier for the reader to know what the current topic is.

- L433: You hypothesized this? I may have missed it in the introduction, but it seems that it wasn’t clear enough then. Or is this a new hypothesis that you bring up here to explain why your primary hypothesis was not confirmed? If so, it belongs into the introduction, backed by a reasoning.

- L444: checking with the matched control group is not a clean test. A power analysis relies on an assumed effect size. It is conceivable that the effect is smaller in MA than in the control and thus cannot be found. I don’t think that is the case in your study, but please don’t present the comparison as if it would be a reliable statistical test.

Discussion:

- L477: neither used real music clips.

- L477ff: I’m missing an interpretation of the results. There is only the single sentence in L485ff, but there is for sure more to be said.

- L491: what are groove stimuli?

- L495: groove is maintained through urge to move: this is circular. You have not provided an understanding of groove that exceeds the pleasurable urge to move. Hence, for someone with MA, we would, in theory, assume that either groove equals the urge to move alone, or they cannot experience groove at all. I understand that your results change this theory, but this needs to be unpacked.

- L507: the perception of perception is also circular.

Conclusion:

- L531: both studies emphasize the urge to move in their definitions of groove, not actual movement. However, there are studies that showed that entrained body movement increases pleasure, e.g., Bernardi et al. (2017).

Mentioned Literature:

Senn, O., Bechtold, T., Rose, D., Câmara, G. S., Düvel, N., Jerjen, R., Kilchenmann, L., Hoesl, F., Baldassarre, A., & Alessandri, E. (2020). Experience of Groove Questionnaire: Instrument Development and Initial Validation. Music Perception, 38(1), 46–65. https://doi.org/10.1525/mp.2020.38.1.46

Senn, O., Bechtold, T., Hoesl, F., Jerjen, R., Kilchenmann, L., Rose, D., Baldassarre, A., Sigrist, C., & Alessandri, E. (2023). An SEM approach to validating the psychological model of musical groove. Journal of Experimental Psychology: Human Perception and Performance, 49(3), 290–305. https://doi.org/10.1037/xhp0001087

Senn, O., Rose, D., Bechtold, T., Kilchenmann, L., Hoesl, F., Jerjen, R., Baldassarre, A., & Alessandri, E. (2019). Preliminaries to a Psychological Model of Musical Groove. Frontiers in Psychology, 10(1228), 1–5. https://doi.org/10.3389/fpsyg.2019.01228

Düvel, N., Labonde, P., Bechtold, T., Senn, O., & Kopiez, R. (2021). Experience of Groove Questionnaire: German Translation and Validation. Music Perception, 39(1), 83–99. https://doi.org/10.1525/mp.2021.39.1.83

Danielsen, A. (2006). Presence and pleasure: The funk grooves of James Brown and parliament. Wesleyan University Press.

Roholt, T. C. (2014). Groove: A Phenomenology of Rhythmic Nuance. Bloomsbury Publishing USA.

Janata, P., Tomic, S. T., & Haberman, J. M. (2012). Sensorimotor coupling in music and the psychology of the groove. Journal of Experimental Psychology. General, 141(1), 54–75. https://doi.org/10.1037/a0024208

Bechtold, T. A., Kilchenmann, L., Curry, B., & Witek, M. A. G. (2023). Understanding the Relationship Between Catchiness and Groove: A Qualitative Study with Popular Music Creators. Music Perception, 40(5), 353–372. https://doi.org/10.1525/mp.2023.40.5.353

Senn, O., Kilchenmann, L., Bechtold, T., & Hoesl, F. (2018). Groove in drum patterns as a function of both rhythmic properties and listeners’ attitudes. PLOS ONE, 13(6), e0199604. https://doi.org/10.1371/journal.pone.0199604

Senn, O., Hoesl, F., Bechtold, T. A., Kilchenmann, L., Jerjen, R., & Witek, M. (2024). Null effect of perceived drum pattern complexity on the experience of groove. OSF. https://doi.org/10.31234/osf.io/5rfw7

Witek, M. A. G., Clarke, E. F., Wallentin, M., Kringelbach, M. L., & Vuust, P. (2014). Syncopation, Body-Movement and Pleasure in Groove Music. PLoS ONE, 9(4), e94446. https://doi.org/10.1371/journal.pone.0094446

Bernardi, N. F., Bellemare-Pepin, A., & Peretz, I. (2017). Enhancement of Pleasure during Spontaneous Dance. Frontiers in Human Neuroscience, 11, 1–14. Scans, Ordner. https://doi.org/10.3389/fnhum.2017.00572

Reviewer #2: Your study investigates how anhedonics respond to groove. Your findings--that the pleasurable urge to move in your anhedonic cohort was similar to that of the control group--is revealing and thought-provoking. I have a few methodological concerns: the sound samples are short; and the feeling of urge to move and pleasure is self-reported. I think you need to offer more justification for why this is an appropriate protocol. For instance, is the urge to move instantaneous--or can it sometimes arise after a longer context has been established? Wouldn't the urge to move be more accurately assessed by directly observing the participants?

In addition, you only offer one musical example, showing a repeated, syncopated dominant 7th chord. Absent a richer musical context, I'm skeptical about pleasure ratings for harmony based on a single chord--this seems simplistic to me. Also, the example is notated poorly rhythmically: in a syncopated rhythm, you should always show where the beat lands: 8th, 16th rest, 16th tied a 16th (to show the beat), 16th rest, etc. I think your Appendix should include more musical examples, since they are a cornerstone of the study.

As a musician, I also found it very difficult to follow how you reported the results: the information is no doubt all there, but it is a puzzle trying to sort through it. For instance, given what you describe, you would expect anhedonics to be indifferent to changes in harmony--since you assert that harmonic complexity influences the urge to move indirectly through its relationship with pleasure, to which anhedonics are presumably indifferent. This seems like an obvious question to me, but I had to read your results several times before I found the answer. I would do more to highlight this finding. Also, in line 197, you mention that the BMRQ evaluates sensory-motor reward responses. Given your findings, do anhedonics report higher ratings there than for other sub-scales? If not, how do you explain that? I have a similar question about your statement in line 472. The majority of popular music is groove oriented. If "the desire to move may itself generate pleasure," wouldn't anhedonics generally enjoy popular music? If they don't, how do you square that with your findings?

Overall, I think you should establish more convincingly how your experimental protocol reflects real world musical experiences and, given the inter-disciplinary topic, more clearly explain your results.

A few small comments:

Line 91 - this should read "altering the content of the chords" (tonality refers to the key, which isn't what you mean).

Line 177- I don't know what you mean by "an extension"--in music, that refers to lengthening a phrase, which isn't the case here.

Line 248: Using the word "difference" and "differ" in the same sentence is confusing. I would rewrite this as "The tests differ in the way the oddball examples are formed" or something like that.

Reviewer #3: sharpen the data analysis and discussion of this research, pay attention to the level of relevance of the method, problems can be sharpened again so that the findings and impact of this research are very large

6. PLOS authors have the option to publish the peer review history of their article (what does this mean?). If published, this will include your full peer review and any attached files.

Reviewer #1: No

Reviewer #2: **Yes: **Anthony K. Brandt

Reviewer #3: No

---

## [Author Response · Author response to Decision Letter 0]

22 Aug 2024

Journal Requirement Questions

The article has been updated to meet PLOS ONE’s style requirements.

We have rectified this.

3. Thank you for stating the following financial disclosure: [This research was supported by a grant to VP from the Natural Sciences and Engineering Research Council of Canada (NSERC 2021-04026). IR was funded by the Canadian Graduate Scholarship - Masters by NSERC and by the Fonds de Rechereche du Québec - Nature et Technologies (FRQ - NT). SDB and NF received funding from a Discovery Grant (RGPIN-2019-05453) from NSERC, and by the Canada Research Chair program (CRC in music auditory-motor skill learning and new technologies).]. Please state what role the funders took in the study. If the funders had no role, please state: ""The funders had no role in study design, data collection and analysis, decision to publish, or preparation of the manuscript."" If this statement is not correct you must amend it as needed. Please include this amended Role of Funder statement in your cover letter; we will change the online submission form on your behalf.

4. Thank you for stating the following in the Competing Interests section:[SDB is on the board of the BeatHealth company dedicated to the design and commercialization of technological tools for assessing rhythm capacities such as BAASTA and implementing rhythm-based interventions.]. Please confirm that this does not alter your adherence to all PLOS ONE policies on sharing data and materials, by including the following statement: "This does not alter our adherence to PLOS ONE policies on sharing data and materials.” (as detailed online in our guide for authors http://journals.plos.org/plosone/s/competing-interests). If there are restrictions on sharing of data and/or materials, please state these. Please note that we cannot proceed with consideration of your article until this information has been declared. Please include your updated Competing Interests statement in your cover letter; we will change the online submission form on your behalf.

Please find the updated statement below.

“SDB is on the board of the BeatHealth company dedicated to the design and commercialization of technological tools for assessing rhythm capacities such as BAASTA and implementing rhythm-based interventions. This does not alter our adherence to PLOS ONE policies on sharing data and materials.”

Reviewers' comments:

Reviewer #1: I have mixed feelings about this article. On one hand, it follows a clever idea to separate urge to move and pleasure in groove, and employs a high qualitative research design to do so. On the other hand, the literature review, theoretical underpinning, and discussion are somewhat lacking and there are some issues with the analysis.

We wanted to thank the reviewer for their thorough and constructive comments. We have attempted to incorporate their feedback throughout the manuscript where possible. 

Major issues:

1. I see some issues with how the model and variable selection were performed; it would be worthwhile to do some more testing and update the models if necessary.

a) Variable selection: you conducted forward addition, but as far as I know, backwards elimination or bidirectional addition/elimination is more common. The latter is in any case the more thorough method and should be preferred.

b) I don’t see a test for the model structure. You state that your baseline model had a by-participant intercept, which is reasonable for a start - although you do not provide your theoretical reasoning for it. However, it seems you did not check for other options. It is counterintuitive to expect that the stimuli are sufficiently described by HC and RC, and that a by-stimulus intercept would thus be unnecessary. Similarly, a slope for RC can be expected: why would every participant react with the same strength to the manipulations of RC? I did a quick test for some of the possible configurations with your data, and it seems that a model with (Rhythm | ID) + (1 | MusicName) would be the most efficient model structure (but I might have made a mistake, and I did not check for the groups independently!). I guess this model will run into convergence issues with your MA subset due to the low amount of data. I am also aware that the mediation package is not able to work correctly with slopes. I have not calculated all your models, and can’t say whether different models would change some of your results. Yet, I think it is necessary to check for additional model structures, at least for the by-stimulus intercept. If you choose to ignore better fits with slopes, you can probably find an argumentation for that, as a) it is not uncommon to skip the respective checks, i.e., provide an argumentation that other papers did not use it and why you think this is the correct way to handle it or b) mediation analysis is central to your study and, as far as I know, this gets a bit messy with random slopes, i.e., provide an argumentation based on your intended method and why you think results are not changed by dropping down to your current model.

We wanted to address these points together as similar reasoning was used for both. While bidirectional addition/elimination is generally the most preferred method in exploratory analyses, it is less preferred for hypothesis-driven analyses (Harrell, 2001). Additionally, we chose a forward-addition model instead of a backward-elimination model because of model convergence issues. Because we wanted to replicate the findings of Matthews et al., 2019, we wanted to include harmonic complexity in the model. As you mentioned in your comment, the maximal model is (Rhythm | ID) + (1 | MusicName). Unfortunately, the model, which includes harmonic complexity as a fixed effect, fails to converge. Therefore, to address our hypothesis, we felt it was better to build the model up in a forward addition process to ensure convergence.

For the mediation analyses, as you mentioned in your comment, unfortunately, random slopes do not work properly in the mediation package. Although it could be ideal to incorporate random slopes, we believe that the mediation analysis is a central finding of the paper and thus have included it over the random slopes.

References:

Harrell, F. E. (2001). Regression modeling strategies: with applications to linear models, logistic regression, and survival analysis (Vol. 608). New York: springer.

Matthews, T. E., Witek, M. A., Heggli, O. A., Penhune, V. B., & Vuust, P. (2019). The sensation of groove is affected by the interaction of rhythmic and harmonic complexity. PLoS One, 14(1), e0204539.

2. The paper is at times very repetitive. In some cases, you are repeating half sentences that you just brought up one or two sentences above. In others, full sentences are almost duplicates. It would enhance the readability of your paper if you could cut down on repetitions.

We have reviewed the manuscript and attempted to eliminate repetitions.

3. Your literature review is one-sided (even more emphasized by frequent mentions of the own lab) and misses some studies that examined the relationship of pleasure and urge to move that should be taken into account. For example, Senn et al. (2020) tried to separate urge to move and pleasure through a specific set of real music stimuli but failed, probably due to methodological problems. In Senn et al. (2023), the correlation between pleasure and urge to move was rather low and pleasure hardly predicted any variance in urge to move ratings, again using stimuli selected to show independence. And probably of most interest to your study is the psychological model of groove (Senn et al., 2019; later adapted) that postulates a so far untested for hedonistic feedback loop between urge to move and pleasure in groove. Their reasoning and argumentation could be quite useful for your introduction and discussion, especially as your study seems to confirm this loop/the half that has not been tested before. In my opinion, your study would benefit greatly from considering these results – and not including some of these leaves the impression that of a hastily literature review process.

Thank you for bringing these articles to our attention. We have now incorporated them into the Introduction. In particular, Senn’s 2019 model of groove fits well alongside our findings, and as such, we have also included it in the Discussion section.

a) You are not sufficiently introducing the idea of a non-separable urge to move and pleasure. On what basis would you think that they would be either the same thing or fuse to a single entity? In case it is because of the high correlation, then an r = 0.8 means that pleasure still cannot explain 36% of the variance in urge to move (and this amount likely increases if you add a random intercept). Hence, I don’t think this is sufficient as an explanation. And besides this strong correlation, Senn et al. (2020) and Düvel et al. (2021) found that the best fitting CFA on their 6-item questionnaire on groove has in fact two constructs (pleasure and urge to move), and not one. This is quite a good argument that they are separable already.

We have now expanded our discussion including, Senn, 2020 in this section and expanded the discussion of the possible separability of pleasure and the urge to move in the sensation of groove.

b) Pleasure is not sufficiently introduced in the introduction nor discussed later. You provide some hints that you subscribe to the idea that multiple pleasures exist or at least pleasure derived from multiple sources. Yet, as this is quite important for your main result and conclusion, this is underdeveloped and lacks proper sources about the nature of pleasure. I’m not aware of a published study that investigates the nature of pleasure in groove in particular; there are some things in Danielsen (2006), Roholt (2014), Janata et al. (2012) and Bechtold et al. (2023) but the last two seemingly preferred the term “positive affect” and I’m not sure if it fits completely with your concept of pleasure. However, there are countless texts on pleasure from many disciplines and your paper would benefit from being clearer on how you actually understand one of the examined core concepts.

We agree with the reviewer that pleasure is a complex topic, and that it likely originates from various sources. In the section on predictive coding (i.e., L81-98), we have expanded on the idea that pleasure from groove originates from the processes of learning and refining of internal models. Although there are other ways to define and understand pleasure, this is the conceptualization that motivated the current study.

c) Musical Anhedonia is also not properly introduced. As a reader, I’m sufficiently informed how you operationalized it in the study, but what it actually is, how it comes about, what body or brain parts it affects, what consequences it has for affected people, etc. is not discussed at all.

Music specific anhedonia has been observed in the general population (congenital anhedonia) and in people with brain injury (acquired anhedonia). Currently, there is no information about genetic contributions to this trait. The very few neuropsychological studies of acquired anhedonia have found no consistently overlapping brain regions, with auditory, parietal, insula and amygdala known to be damaged in individual cases (Belfi & Loui, 2020). A brain imaging study of congenital amusia showed reduced activity in auditory and reward regions, along with decreased white matter connectivity between the primary auditory cortex and reward networks (Martínez-Molina et al., 2016). This additional information has been added to the Introduction on lines 130-133.

References:

Belfi, A. M., & Loui, P. (2020). Musical anhedonia and rewards of music listening: Current advances and a proposed model. Annals of the New York Academy of Sciences, 1464(1), 99-114.

Martínez-Molina, N., Mas-Herrero, E., Rodríguez-Fornells, A., Zatorre, R. J., & Marco-Pallarés, J. (2016). Neural correlates of specific musical anhedonia. Proceedings of the National Academy of Sciences, 113(46), E7337-E7345.

4. The method section is very long and, in my opinion, not properly organized. You start with a pseudo-procedure, then go to participants which features many things that have not been introduced before. Then you have a section about rating groove that mostly discusses the stimuli. Afterwards, there is a very long section about measures, followed by a very short section about measures. Then, you have a proper procedure and a data analysis section. My suggestion to make this more streamlined: drop the initial procedure, and start with a section about all measures, then stimuli, then a proper procedure and last the analysis. And regarding the measures, I think you are giving way too many details that are not particularly interesting to the reader. Their main purpose seems to be to check whether a participant has MA and are consequently not discussed much in the results or discussions. I would suggest to significantly cut down on this (e.g., no need to report the exact procedure, stimuli, nor analysis for these if they are established tests) and put the rest in a supplementary file.

We agree that the Methods section could have been better organized. Based on the reviewer’s suggestions, we have revised and shortened it. 

Minor issues:

Title:

I’m not a native speaker, but to me the title sounds incomplete. Unchanged from what? And is “in Musical Anhedonia” correct? Shouldn’t it be “in People with Musical Anhedonia” or something similar.

We have revised the title to: The Pleasurable Urge to Move to Music is Unchanged in People with Musical Anhedonia

Abstract:

The abstract is too long and detailed. It should serve as a quick overview over the study and would benefit from being shorter and the main takeaways presented earlier.

The Abstract conforms to the length guidelines of the journal. As part of our revisions, we have made some changes to improve readability. 

o L38: I do not think that N=148 qualifies as large and you do not lose anything by cutting this claim. Especially, N = 17 is not large. 

The word “large” was omitted from the abstract.

o L38/L44: you are describing your stimuli as rhythmic but then later they apparently have harmonic complexity. It is clear in the paper but incomprehensible in the abstract. 

We have altered how we describe our stimuli in order to be more explicit that they vary in both rhythmic and harmonic complexity.

o L47: “response” is very technical, how about “sensation” or “experience”? 

“Response” has been changed to “sensation”.

Introduction:

- L51: “often generates a…” is too vague.

We have altered the sentence to say “…of listening to music is that it is often accompanied by a pleasurable urge to move along.”

- L64: please provide the citations after each structural component. 

Relevant citations have been included after each structural component of groove is mentioned.

- L71: predictive coding and groove: this theory is en vogue and it might have some importance to your study and thus warrant inclusion. However, the way you summed it up, it does not provide a useful explanation for how groove comes about. If I take just your information, it sounds like I cannot ever experience groove with music where my prediction error is very low (e.g., because I know it very well). This is counterintuit

---

## [Decision Letter · Decision Letter 1]

22 Sep 2024

PONE-D-24-06715R1The pleasurable urge to move to music is unchanged in people with musical anhedoniaPLOS ONE

Dear Dr. Romkey,

Thank you for submitting your manuscript to PLOS ONE. After careful consideration, we feel that it has merit but does not fully meet PLOS ONE’s publication criteria as it currently stands. Therefore, we invite you to submit a revised version of the manuscript that addresses the points raised during the review process. 

We look forward to receiving your revised manuscript.

Kind regards,

Hira Rafi

Academic Editor

PLOS ONE

Journal Requirements:

Reviewers' comments:

Reviewer's Responses to Questions

**Comments to the Author**

1. If the authors have adequately addressed your comments raised in a previous round of review and you feel that this manuscript is now acceptable for publication, you may indicate that here to bypass the “Comments to the Author” section, enter your conflict of interest statement in the “Confidential to Editor” section, and submit your "Accept" recommendation.

Reviewer #1: (No Response)

Reviewer #2: (No Response)

Reviewer #3: All comments have been addressed

2. Is the manuscript technically sound, and do the data support the conclusions?

Reviewer #1: Yes

Reviewer #2: Partly

Reviewer #3: Yes

3. Has the statistical analysis been performed appropriately and rigorously? 

Reviewer #1: Yes

Reviewer #2: I Don't Know

Reviewer #3: Yes

4. Have the authors made all data underlying the findings in their manuscript fully available?

Reviewer #1: Yes

Reviewer #2: Yes

Reviewer #3: Yes

5. Is the manuscript presented in an intelligible fashion and written in standard English?

Reviewer #1: Yes

Reviewer #2: Yes

Reviewer #3: Yes

6. Review Comments to the Author

Reviewer #1: I would like to thank the authors for their careful revision. All my points were addressed (except for one thing, see below) and I think the manuscript is in a much better state in its current form.

The one issue I still have is transparency with modelling decision. I can fully comprehend that you selected the variables based on previous studies, and that you dismissed slopes and an additional intercept as you wanted to stick with the mediation package. Yet, I think it needs to be stated in the paper as well, not just hidden in the answer to a reviewer. Similarly to what I said in my original comment: you can add a statement that you preselected the variables based on theory/desire to replicate previous findings and did not check for other options or combinations. And you can make the argument that the package (and low amount of MA participant) prohibits using slopes/MusicName intercept. But, as I said, this needs to be made clear to the reader, not just the reviewer.

I trust the authors and editor that this comment is incorporated or dismissed without the need for another round of reviews.

---

On a personal note, I think that convergence issues shouldn’t play a role in study with sufficient funds in the background and universities that have staff that can provide help with statistics. They can usually be avoided through better modelling, changing the statistical framework, stronger experimental design, or having more participants. Simulating the statistical power beforehand can give you an indication whether such changes to your plans are necessary. But sometimes, the actual data differs from expectations and the analysis must adapt accordingly. It is too late for this study, but for future work, it might be beneficial to consult with a statistician what options you have with your data specifically. In my eyes, it’s has been a malpractice for the past years in music psychology to not test more rigorously for model structure. I can only suspect that convenience (i.e. avoiding such convergence issues) or a learned disregard due to the often small changes in results (until you find yourself with data in which slopes are crucial for a correct picture) are the motivations behind this. As a minor note: (Rhythm | ID) + (1 | MusicName) is not at all the maximal structure, it is what I found to be the most efficient one in a quick test. The maximum would be along the lines of (Rhythm * Harmony | ID) + (Rhythm * Harmony | MusicName).

Reviewer #2: Thank you for addressing many of questions raised in the initial review. I still have concerns about the methodology, and feel you should include some evidence that urge to move is evident within 10 seconds and remains constant when the duration is longer--i.e. situations in which the urge to move would only arise after 10 secs for an unchanging stimulation are rare/unlikely. I also think it would be worth justifying using subjective measures to measure the urge to move, as opposed to direct observation. For instance, have previous studies shown a high correlation between the two?

There are several typos:

line 70: wanting to move

86: musical structure

92: Meanwhile, in complex rhythms (I don't think you can start a sentence with "While")

93: the inverse is true: ... (colon, not comma)

199: check grammar here

258: awkward wording - I would say "Lastly, high-complexity added a flat ninth above the root of the chord."

262: FIGURE 1 is not a four-note chord: it is a ninth chord D-F#-A-C-E. This is a rather concerning mistake. Is it misnotated? Or is this the actual chord that was played for participants?

For comprehensiveness, I would like to see all--or at least a strong sampling--of musical examples included in the Appendix. Including links to sound samples would be even better, so the reader can test the various stimuli themselves.

Finally, I have a question about "the dampening of overall pleasure" discussed in line 486. For the control group, "pleasure" in these samples is presumably due to several causes, whereas--as you explain--for the anhedonic group, it only comes from the urge to move. Is it possible that this explains the "dampening of overall pleasure"? I.e. Both the control group and anhedonics get the same amount of pleasure from the urge to move; but the control group adds additional sources of pleasure? Or am I misunderstanding in some way?

Reviewer #3: The author has at least made changes to the manuscript in detail and thoroughly, the use of language also meets writing standards. The author has fulfilled the completeness of supporting data

7. PLOS authors have the option to publish the peer review history of their article (what does this mean?). If published, this will include your full peer review and any attached files.

Reviewer #1: No

Reviewer #2: **Yes: **Anthony K. Brandt

Reviewer #3: **Yes: **Hery Budiawan, Art Education, Universitas Negeri Jakarta

---

## [Author Response · Author response to Decision Letter 1]

25 Sep 2024

Reviewer #1: I would like to thank the authors for their careful revision. All my points were addressed (except for one thing, see below) and I think the manuscript is in a much better state in its current form.

The one issue I still have is transparency with modelling decision. I can fully comprehend that you selected the variables based on previous studies, and that you dismissed slopes and an additional intercept as you wanted to stick with the mediation package. Yet, I think it needs to be stated in the paper as well, not just hidden in the answer to a reviewer. Similarly to what I said in my original comment: you can add a statement that you preselected the variables based on theory/desire to replicate previous findings and did not check for other options or combinations. And you can make the argument that the package (and low amount of MA participant) prohibits using slopes/MusicName intercept. But, as I said, this needs to be made clear to the reader, not just the reviewer.

I trust the authors and editor that this comment is incorporated or dismissed without the need for another round of reviews.

---

On a personal note, I think that convergence issues shouldn’t play a role in study with sufficient funds in the background and universities that have staff that can provide help with statistics. They can usually be avoided through better modelling, changing the statistical framework, stronger experimental design, or having more participants. Simulating the statistical power beforehand can give you an indication whether such changes to your plans are necessary. But sometimes, the actual data differs from expectations and the analysis must adapt accordingly. It is too late for this study, but for future work, it might be beneficial to consult with a statistician what options you have with your data specifically. In my eyes, it’s has been a malpractice for the past years in music psychology to not test more rigorously for model structure. I can only suspect that convenience (i.e. avoiding such convergence issues) or a learned disregard due to the often small changes in results (until you find yourself with data in which slopes are crucial for a correct picture) are the motivations behind this. As a minor note: (Rhythm | ID) + (1 | MusicName) is not at all the maximal structure, it is what I found to be the most efficient one in a quick test. The maximum would be along the lines of (Rhythm * Harmony | ID) + (Rhythm * Harmony | MusicName).

We want to thank reviewer #1 for their comments. We have added sentences (L310 - 314) to be completely transparent regarding our decisions about our models. Additionally, we wanted to thank the reviewer for their personal note, and we will consider this advice for future studies.

Reviewer #2: Thank you for addressing many of questions raised in the initial review. I still have concerns about the methodology, and feel you should include some evidence that urge to move is evident within 10 seconds and remains constant when the duration is longer--i.e. situations in which the urge to move would only arise after 10 secs for an unchanging stimulation are rare/unlikely.

Previous studies have utilized samples of a variety of lengths, both longer and shorter than 10 seconds, and elicited groove (Leow et al., 2014; Spiech et al., 2022 [Preprint]; Witek et al., 2014. The other component of your question, if the urge to move remains constant over an extended period of time, has yet to be tested. This would be an interesting question to investigate in future studies but is outside the scope of the present study.

References:

Leow, L. A., Parrott, T., & Grahn, J. A. (2014). Individual differences in beat perception affect gait responses to low-and high-groove music. Frontiers in human neuroscience, 8, 811.

Spiech, C., Hope, M., Câmara, G. S., Sioros, G., Endestad, T., Laeng, B., & Danielsen, A. (2022, July 26). Sensorimotor Synchronization Increases Groove. https://doi.org/10.31234/osf.io/fw7mh

Witek, M. A., Clarke, E. F., Wallentin, M., Kringelbach, M. L., & Vuust, P. (2014). Syncopation, body-movement and pleasure in groove music. PloS one, 9(4), e94446.

 I also think it would be worth justifying using subjective measures to measure the urge to move, as opposed to direct observation. For instance, have previous studies shown a high correlation between the two?

As stated in our previous response to this question, for this study, we are focusing specifically on the urge to move, not movement itself. While previous work has found that the two are related (Janata et al., 2012; Spiech et al., 2022), an individual may experience an urge to move to the music without physically moving to the music at all. By its very nature, the urge to move is a subjective experience, and as such, it must be assessed via subjective measures. Future work could investigate the role movement may play in this relationship, but this part is outside the scope of the current study.

References:

Janata, P., Tomic, S. T., & Haberman, J. M. (2012). Sensorimotor coupling in music and the psychology of the groove. Journal of experimental psychology: General, 141(1), 54.

Spiech, C., Hope, M., Câmara, G. S., Sioros, G., Endestad, T., Laeng, B., & Danielsen, A. (2022, July 26). Sensorimotor Synchronization Increases Groove. https://doi.org/10.31234/osf.io/fw7mh

There are several typos:

line 70: wanting to move

This has been corrected.

86: musical structure

This has been addressed.

92: Meanwhile, in complex rhythms (I don't think you can start a sentence with "While")

We have altered the beginning of the sentence to improve readability.

93: the inverse is true: ... (colon, not comma)

The comma has been replaced by a colon.

199: check grammar here

This has been addressed.

258: awkward wording - I would say "Lastly, high-complexity added a flat ninth above the root of the chord."

We have altered the sentence per your suggestion.

262: FIGURE 1 is not a four-note chord: it is a ninth chord D-F#-A-C-E. This is a rather concerning mistake. Is it misnotated? Or is this the actual chord that was played for participants?

We think this is an issue of terminology. This the actual chord that was played to the participants. We used the term “four-note chords with extensions” as the descriptor for the medium complexity chords used in our study. The stimuli consisted of various four-note D major chords with an additional fifth note, or an ‘extension.’ In this case, the term ‘extension’ comes from jazz theory (Gross. 2015) and refers to a note that goes above the octave (e.g., a 9th or a 13th). So, the chord illustrated in figure 1 would be an example of a four-note chord with an extension; in particular, it would be a ninth chord.

We have adjusted the figure caption to coincide better with how these stimuli were used in previous work (Matthews et al., 2019; Witek et al., 2023).

References:

Gross, A. (2015). Jazz Theory: From Basic to Advanced Study.

Matthews, T. E., Witek, M. A., Heggli, O. A., Penhune, V. B., & Vuust, P. (2019). The sensation of groove is affected by the interaction of rhythmic and harmonic complexity. PLoS One, 14(1), e0204539.

Witek, M. A., Matthews, T., Bodak, R., Blausz, M. W., Penhune, V., & Vuust, P. (2023). Musicians and non-musicians show different preference profiles for single chords of varying harmonic complexity. Plos One, 18(2), e0281057.

For comprehensiveness, I would like to see all--or at least a strong sampling--of musical examples included in the Appendix. Including links to sound samples would be even better, so the reader can test the various stimuli themselves.

We have included a folder in the supplementary materials so that individuals can access all of the sound samples used in our study.

Finally, I have a question about "the dampening of overall pleasure" discussed in line 486. For the control group, "pleasure" in these samples is presumably due to several causes, whereas--as you explain--for the anhedonic group, it only comes from the urge to move. Is it possible that this explains the "dampening of overall pleasure"? I.e. Both the control group and anhedonics get the same amount of pleasure from the urge to move; but the control group adds additional sources of pleasure? Or am I misunderstanding in some way?

We agree with the reviewer’s understanding of our interpretation of these findings.

Reviewer #3: The author has at least made changes to the manuscript in detail and thoroughly, the use of language also meets writing standards. The author has fulfilled the completeness of supporting data

We thank reviewer #3 for their comments.

---

## [Decision Letter · Decision Letter 2]

30 Sep 2024

The pleasurable urge to move to music is unchanged in people with musical anhedonia

PONE-D-24-06715R2

Dear Dr. Romkey,

We’re pleased to inform you that your manuscript has been judged scientifically suitable for publication and will be formally accepted for publication once it meets all outstanding technical requirements.

Kind regards,

Hira Rafi

Academic Editor

PLOS ONE

Additional Editor Comments (optional):

Reviewers' comments:

Reviewer's Responses to Questions

**Comments to the Author**

1. If the authors have adequately addressed your comments raised in a previous round of review and you feel that this manuscript is now acceptable for publication, you may indicate that here to bypass the “Comments to the Author” section, enter your conflict of interest statement in the “Confidential to Editor” section, and submit your "Accept" recommendation.

Reviewer #2: (No Response)

2. Is the manuscript technically sound, and do the data support the conclusions?

Reviewer #2: Yes

3. Has the statistical analysis been performed appropriately and rigorously? 

Reviewer #2: I Don't Know

4. Have the authors made all data underlying the findings in their manuscript fully available?

Reviewer #2: Yes

5. Is the manuscript presented in an intelligible fashion and written in standard English?

Reviewer #2: Yes

6. Review Comments to the Author

Reviewer #2: For readers not familiar with jazz theory, "four-note chord extended chords" is likely to be confusing when the chord has five notes. It's also confusing when you say "for high-complexity we added a flat ninth"--which sounds like there are more chord tones than in the medium-complexity one. For the medium-complexity, I would say "dominant ninth chord" or "dominant seventh plus a ninth" and for high-complexity, "we flatted the ninth above the root of the chord, making it more dissonant" (you can leave off the comment about dissonance if you prefer).

There's no need for an additional review after you make this correction. Congratulations on publishing the paper!

7. PLOS authors have the option to publish the peer review history of their article (what does this mean?). If published, this will include your full peer review and any attached files.

Reviewer #2: **Yes: **Anthony K Brandt

---

## [Editor Report · Acceptance letter]

7 Nov 2024

PONE-D-24-06715R2 

PLOS ONE

Dear Dr. Romkey, 

I'm pleased to inform you that your manuscript has been deemed suitable for publication in PLOS ONE. Congratulations! Your manuscript is now being handed over to our production team.

Kind regards, 

on behalf of

Dr. Hira Rafi 

Academic Editor

PLOS ONE